# Radiative cooling assisted self-sustaining and highly efficient moisture energy harvesting

Chenyue Guo[1,5], Huajie Tang[1,5], Pengfei Wang [2,5], Qihao Xu[1], Haodan Pan[1], Xinyu Zhao[1], Fan Fan[1], Tingxian Li [2] ✉ & Dongliang Zhao [1,3,4] ✉

Harvesting electricity from ubiquitous water vapor represents a promising route to alleviate the energy crisis. However, existing studies rarely comprehensively consider the impact of natural environmental fluctuations on electrical output. Here, we demonstrate a bilayer polymer enabling self-sustaining and highly efficient moisture-electric generation from the hydrological cycle by establishing a stable internal directed water/ion flow through thermal exchange with the ambient environment. Specifically, the radiative cooling effect of the hydrophobic top layer prevents the excessive daytime evaporation from solar absorption while accelerating nighttime moisture sorption. The introduction of LiCl into the bottom hygroscopic ionic hydrogel enhances moisture sorption capacity and facilitates ion transport, thus ensuring efficient energy conversion. A single device unit (1 cm²) can continuously generate a voltage of ~0.88 V and a current of ~306 µA, delivering a maximum power density of ~51 µW cm⁻² at 25 °C and 70% relative humidity (RH). The device has been demonstrated to operate steadily outdoors for continuous 6 days.

Growing energy crisis calls for an urgent need to develop eco-friendly energy sources. Water vapor, an indispensable recyclable resource, serves as the largest energy carrier on the Earth. A significant portion of solar energy reaching the Earth's surface has been stored, transmitted, and converted through the hydrological cycle[1–3]. However, energy occurring in this process remains largely untapped[4]. Propelled by rapid advancements in nanoscience, harvesting energy directly from the ubiquitous moisture has demonstrated potential[5–7]. Moisture-electric generators (MEGs), capitalizing on the spontaneous interaction between moisture and functionalized nanomaterials, can convert chemical potential energy released by the state transition of water molecules into electrical power[8,9]. In previous studies, materials represented by graphene oxide were first demonstrated for intermittent electricity generation by constructed oxygen-containing group gradient[5,10,11]. Subsequently, continuous power output has been devised by establishing a self-maintained moisture gradient in materials such as polyelectrolyte films[12] and protein nanowires[13].

Current research focuses on enhancing the power output performance of MEGs through material regulation and structural optimization[14–18]. However, the gradual saturation of moisture content during the single sorption process leads to an inevitable decrease in ionic concentration gradient and subsequent current decay. To address this challenge, some studies have integrated moisture sorption and desorption processes, utilizing the driving force of ion-hydration energy during desorption[19] or employing a simultaneous sorption-evaporation working mode[20,21]. Existing research primarily focuses on sorption/desorption induced by humidity variations, while fluctuations in solar radiation and ambient temperature during day-night transitions also show potential for promoting the hydrological cycle in real-world application. Systematically considering the influence of natural environmental factors on the performance of MEGs is necessary.

Daytime radiative cooling achieves passive cooling without energy consumption during the whole diurnal cycle by reflecting sunlight and emitting mid-infrared thermal radiation to the space[22–26].

[1]School of Energy and Environment, Southeast University, Nanjing, China. [2]Institute of Refrigeration and Cryogenics, School of Mechanical Engineering, Shanghai Jiao Tong University, Shanghai, China. [3]Institute of Science and Technology for Carbon Neutrality, Southeast University, Nanjing, China. [4]Institute for Carbon Neutral Development, Southeast University, Nanjing, China. [5]These authors contributed equally: Chenyue Guo, Huajie Tang, Pengfei Wang. ✉e-mail: litx@sjtu.edu.cn; dongliang_zhao@seu.edu.cn

Recent studies have integrated radiative cooling with various sustainable electricity harvesting technologies. For example, radiative cooling film-based triboelectric nanogenerators have been used for personal thermal management and biomechanical energy harvesting[27,28]. Radiative cooling has also been employed to increase the temperature difference between the hot and cold sides of thermoelectric generators, thereby improving power output performance under solar irradiation[29,30]. Additionally, an asymmetric bilayer cellulose-based fabric that is capable of independent radiative cooling and transpiration-driven electricity generation has been developed[31]. In this work, we aim to leverage the influence of radiative cooling process on moisture sorption-desorption kinetics and propose an efficient and sustainable moisture-electric generation solution from a thermodynamic perspective by leveraging the external environment. By strategically introducing radiative cooling to improve thermal management between materials and the environment, it holds promise for constructing a long-term hydrological cycle to maintain directed water/ion flow within the material under real-world environmental fluctuations.

In pursuit of this concept, we developed a bilayer polymer composed of a hydrophobic porous poly(vinylidene fluoride-co-hexafluoropropene) [P(VdF-HFP)] layer and a hygroscopic ionic hydrogel layer (PP/IH), enabling high-efficiency and self-sustaining moisture-electric generation in outdoor environment. The bottom hydrogel layer spontaneously sorbs moisture and dissociates ions, while the transported moisture is exhaled through the porous P(VdF-HFP) top layer. The dynamic equilibrium of sorption-evaporation drives continuous water/ion flow through negatively charged nanochannels, resulting in sustained power output[32]. The radiative cooling effect from the porous P(VdF-HFP) contributes to prolonging the evaporation process during the day, maintaining water flow within the hydrogel, and increasing local humidity at night to accelerate moisture sorption[33–35]. The ionic hydrogel network, established by poly(vinyl alcohol) (PVA) and phytic acid (PA), in conjunction with LiCl which

weakens hydrogen-bond interactions, imparts excellent moisture sorption capability, plentiful dissociated charge carriers, and efficient water/ion transport performance to the PP/IH. The resultant device delivers stable electrical output for over 6 days during outdoor experiments. A single PP/IH unit (1 cm$^2$) can produce a voltage of ~0.88 V and a current of ~306 µA, with a maximum output power density of approximately 51 µW cm$^{-2}$ at 25 °C and 70% RH. This work provides insights for the development of sustainable and efficient MEGs.

## Results

### Design principle and characterization of the PP/IH

As shown in Fig. 1a, the PP/IH consists of a hydrophobic porous P(VdF-HFP) top layer and a hygroscopic ionic hydrogel bottom layer (Supplementary Fig. 1), with carbon cloth electrodes connected to the hydrogel surfaces. The abundant hydrophilic functional groups (Supplementary Fig. 2) and strong hygroscopicity of LiCl enable the hydrogel to spontaneously harvest moisture from the atmosphere and dissociate ions. The plentiful nanochannels within the hydrogel (Supplementary Fig. 3) provide pathways for the directed transport of water and ions. Moreover, the hierarchical pores of the top layer maintain hydrophobicity while permitting evaporated vapor to pass through. This asymmetric hygroscopic structure enables the PP/IH to establish internal water content and ion concentration gradients, delivering a sustained driving force for the continuous water/ion flow through the sorption-desorption dynamic equilibrium (Fig. 1b). The passive radiative cooling effect of porous P(VdF-HFP) establishes a stable hydrological cycle in fluctuating outdoor environment. Specifically, the porous P(VdF-HFP) prevents inherent solar radiation absorption by water, avoiding the reduction in the moisture gradient and water flow rate within the hydrogel caused by excessive daytime evaporation. At night, the radiative cooling power from the top layer balances the sorption heat of the hydrogel, accelerating water sorption for subsequent evaporation.

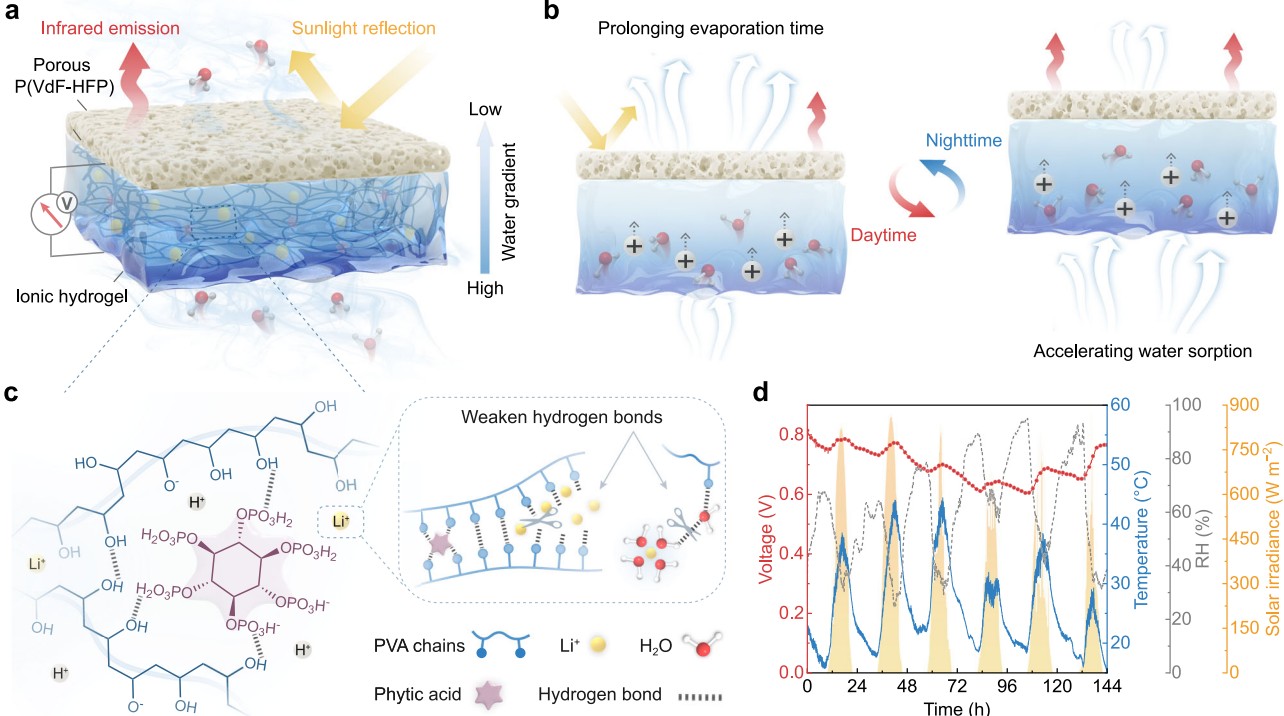

**Fig. 1 | Design of the PP/IH for outdoor self-sustaining moisture-electric generation. a** Schematic diagram illustrating the structure of the PP/IH. **b** The operational mode of the PP/IH in the diurnal cycle. **c** The corresponding material composition and interactions that constitute the ionic hydrogel. **d** A continuous recording of open-circuit voltage from the PP/IH for 6 days in outdoor environment (from 20:20 on April 15 to 20:20 on April 21, 2023, in Nanjing, China).

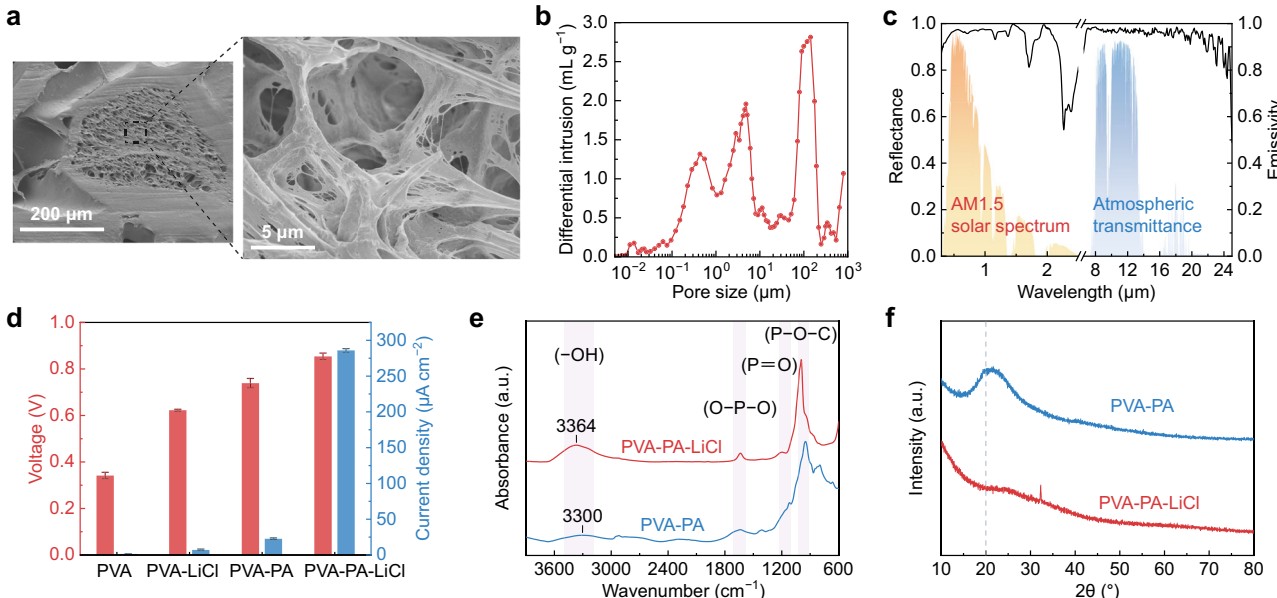

**Fig. 2 | Properties of the porous P(VdF-HFP) and ionic hydrogel. a** The cross-section scanning electron microscope images of the hierarchically porous P(VdF-HFP) film. **b** Pore size distribution of the P(VdF-HFP) film. **c** Solar reflectance (left) and mid-infrared emissivity (right) spectra of a 2-mm-thick porous P(VdF-HFP) film. For reference, the normalized solar irradiance spectrum (orange area) and the atmospheric transmittance (blue area) are also plotted. **d** Comparison of electrical output among hydrogels with different compositions. Error bars represent the standard deviation ($n = 4$). **e** FTIR spectra of PVA-PA and PVA-PA-LiCl hydrogels. **f** XRD patterns of hydrogels with and without LiCl.

The ionic hydrogel was fabricated through a one-pot method by adding PVA and PA into a LiCl aqueous solution and stirring at 95 °C for 3 h, followed by freeze-thaw treatment. PA serves as crosslinking points by reacting with hydroxyl groups on the PVA chains (Supplementary Fig. 2), establishing the polymeric skeleton based on hydrogen bonds (Fig. 1c). Simultaneously, PA with six esterified phosphoric acids dissociates massive mobile H[+] ions when interacting with water molecules[36]. The introduction of inorganic salt (LiCl) weakens hydrogen bonds within the hydrogel, facilitating ion migration[18,37]. Unlike the strategy of soaking synthesized hydrogel in a high-concentration salt solution to tune the aggregation of the polymer chains for improved mechanical and electrical properties[38,39], the direct addition of LiCl can effectively hinder the formation of hydrogen bonds during freeze-thaw process due to the strong hydration of Li[+40]. The obtained ionic hydrogel possesses a loose crosslinked structure, promoting ionic transport rate and providing excellent interface adhesion, enabling firm connection with the electrodes and top layer (Supplementary Fig. 4).

Benefiting from the radiative cooling regulation and the rapid water/ion transport, the PP/IH exhibits superior environmental adaptability. Even in fluctuating outdoor conditions, a consistent water/ion flow can be maintained in the device. Outdoor test results demonstrate that the PP/IH can achieve continuous output for over 6 days under intense solar radiation in hot weather (Fig. 1d). This bilayer structure design fully improves the thermal exchange between materials and the environment, enhancing the energy conversion efficiency in the moisture-electric generation process, thereby achieving self-sustaining and highly efficient outdoor power generation performance.

The scanning electron microscope images in Fig. 2a illustrate the hierarchically porous structure of the top layer. The pore sizes are broadly distributed and concentrated at around 0.4, 4.5, and 150 μm (Fig. 2b). Considering the spectral properties and vapor permeability of the top layer, the thickness of porous P(VdF-HFP) film was chosen to be 2 mm (Supplementary Fig. 5). Owing to the intrinsic electromagnetic characteristics of P(VdF-HFP) and its abundant hierarchical pores, the porous P(VdF-HFP) film demonstrates an average solar reflectivity of ~0.96 and a high thermal emissivity of ~0.97 in the atmospheric transparent window (8–13 μm) (Fig. 2c). Specifically, P(VdF-HFP) exhibits multiple extinction peaks in the atmospheric window due to the diverse vibrational modes of its molecular structure (Supplementary Fig. 6), resulting in strong absorption in this range[41,42]. The abundant nano- and micropores strongly scatter sunlight, while pore size distributed at ~150 μm allow vapor transmission. Additionally, the porous surface and effective medium behavior at large wavelengths facilitate a gradual transition in refractive index across the polymer-air boundary[24,34], leading to a further enhancement in infrared emissivity.

Comparing the electrical output of devices featuring distinct compositions (Fig. 2d, Supplementary Fig. 7), it is obvious that the PA and LiCl synergistically enhances the electrical output. Upon adding LiCl and PA individually, there is an improvement in the voltage output. However, the PVA-PA-LiCl hydrogel exhibits a remarkable increase in both voltage and current. Concretely, PA provides massive migratory ions, resulting in effective charge separation and an increase in voltage output[17]. The introduction of LiCl contributes to elevating the ionic transport rate[43], prompting an obvious increase in current density. Additionally, LiCl significantly enhances the moisture sorption capacity of the hydrogel (Supplementary Fig. 8), enabling it to rapidly establish gradients of water content and ion concentration.

To further investigate the structure and hydrogen-bond interactions within the hydrogels, we performed Fourier transform infrared (FTIR) spectroscopy and X-ray diffraction (XRD) analyses. The FTIR spectra reveal that the O−H stretching absorption peak (3700–3000 cm⁻¹) of PVA-PA hydrogel shifts towards lower wavenumbers (red shift), suggesting a stronger hydrogen-bond interaction compared to PVA-PA-LiCl[17,44] (Fig. 2e). Moreover, from the XRD patterns, a distinct crystallization peak at 20° of PVA-PA can be observed. With the introduction of LiCl, the crystallization peak shifts to a higher degree, simultaneously decreasing in intensity and broadening (Fig. 2f). These observations imply that LiCl disrupts the orderly arrangement of hydrogen bonds in crystalline regions[37,38].

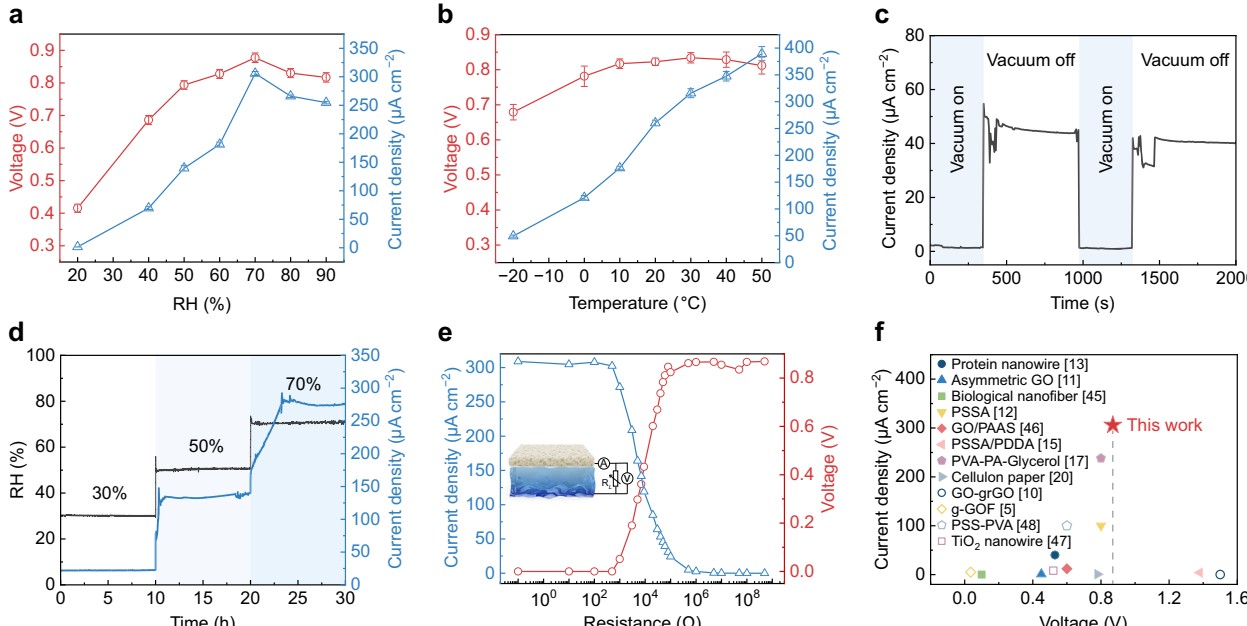

**Fig. 3 | Moisture-electric generation performance of the PP/IH. a** The variation of stable $V_{oc}$ and $I_{sc}$ versus RH change at the temperature of 25 °C. All the samples were dried and short-circuit processed before testing. Error bars represent the standard deviation ($n = 4$). **b** Electrical output in response to temperature at 75% RH. Error bars represent the standard deviation ($n = 4$). **c** The curve of $I_{sc}$ variation in a vacuum-relief alternating environment (23 °C, 40% RH). **d** Evolution of $I_{sc}$ with a step increase in RH. The PP/IH device was dried before testing. **e** Electrical output of the PP/IH with varying external resistances ranging from 0.1 Ω to 500 MΩ at 25 °C and 70% RH ambient conditions. The inset shows a schematic diagram of the equivalent test circuit. **f** The output performances comparison of the PP/IH and the reported moisture-enabled generators[5,10–13,15,17,20,45–48]. The solid dots represent continuous power output, while the hollow dots represent intermittent output.

Additionally, the ionic conductivity of hydrogel with LiCl presenting a substantial improvement compared to PVA-PA (Supplementary Fig. 9). The above observations demonstrate that the strong hydration interaction of Li⁺ inhibits the formation of hydrogen bonds, facilitating rapid ion transport and thereby contributing to the enhanced electrical performance.

## Moisture-electric generation performance of the PP/IH

The electrical output performance of the PP/IH is closely related to ambient temperature and relative humidity. The electrical output test for both open-circuit voltage ($V_{oc}$) and short-circuit current ($I_{sc}$) from 20% to 90% RH shows similar trends, initially increasing and then decreasing (Fig. 3a). Even at a low RH of 20%, the PP/IH maintains a $V_{oc}$ of 0.41 V, reaching a maximum $V_{oc}$ of -0.88 V at 70% RH. $I_{sc}$ exhibits a high sensitivity to RH changes, rapidly increasing from 1.5 μA cm⁻² at 20% RH to 69 μA cm⁻² at 40% RH and achieving a maximum value of 306 μA cm⁻² at 70% RH. The rise in $V_{oc}$ and $I_{sc}$ is primarily attributed to efficient moisture sorption at high RH, promoting dissociation and rapid migration of ions[17]. However, excessive humidity may hinder water evaporation, leading to reduced output. The presence of LiCl in the ionic hydrogel contributes to excellent anti-freezing properties[37] and water retention capability, enabling the PP/IH to function effectively in a broad temperature range from −20 °C to 50 °C (Fig. 3b). Due to enhanced ionic transport at higher temperatures, the $I_{sc}$ gradually increases. These results indicate that both temperature and humidity significantly influence the balance between moisture sorption and evaporation[20], leading to variations in electrical output under diverse environmental conditions. When temperature and humidity are within appropriate ranges, moisture sorption and evaporation can maintain a long-term dynamic balance, enabling continuous and efficient directed water/ion migration in the PP/IH.

From Fig. 3c, d, it is evident that moisture serves as the energy source for the PP/IH. Under vacuum conditions, the device exhibits extremely low current output (Fig. 3c). Upon vacuum release, the current immediately increases. The elevated RH accelerates moisture sorption, resulting in a corresponding gradient increase in $I_{sc}$ (Fig. 3d). Besides, the direction of moisture gradient (Supplementary Fig. 10), as well as the device thickness and size (Supplementary Fig. 11), can affect ion transport behavior, thereby influencing the electrical output performance of the PP/IH. Furthermore, the dependence of voltage and current on the electrical resistance of the external circuit was measured (Fig. 3e). As the resistance increases from 0.1 Ω to 500 MΩ, the voltage gradually increases while the current decreases. The maximum power output of 51.5 μW cm⁻² is achieved at -9 kΩ (Supplementary Fig. 12). Repetitive charge-discharge tests of the PP/IH connected to an external resistor further demonstrate its power output stability (Supplementary Fig. 13). Figure 3f and Supplementary Table 1 compare the output performance of previously reported MEGs[5,10–13,15,17,20,45–48] with the PP/IH in this study. The PP/IH exhibits remarkable electrical output performance, achieving a maximum $V_{oc}$ of 0.88 V and $I_{sc}$ of 306 μA cm⁻², making it promising for practical utilization in self-powered devices. Moreover, compared to generators with intermittent output, the PP/IH provides high environmental adaptability and long-term stable output, further expanding its potential applications.

## Mechanism of electricity generation in the PP/IH

In the context of polymer-based MEGs designed for continuous electrical output, earlier studies have introduced mechanisms centered on proton dissociation and directed transport attributed to moisture sorption[12,17,18]. The output performance of the PP/IH aligns with this principle, as further confirmed in the subsequent experiments. To visually demonstrate the process of moisture diffusion and ion migration, the PP/IH is vertically positioned at the bottom of a beaker containing an isopropanol solution of bromophenol blue pH indicator (Fig. 4a). Upon adding water, the solution on the hydrogel side progressively changes to yellow, indicating H⁺ ions dissociation from the hydrogel. As moisture diffuses, ions migrate toward the porous P(VdF-HFP) side, ultimately causing both sides of the solution turning yellow.

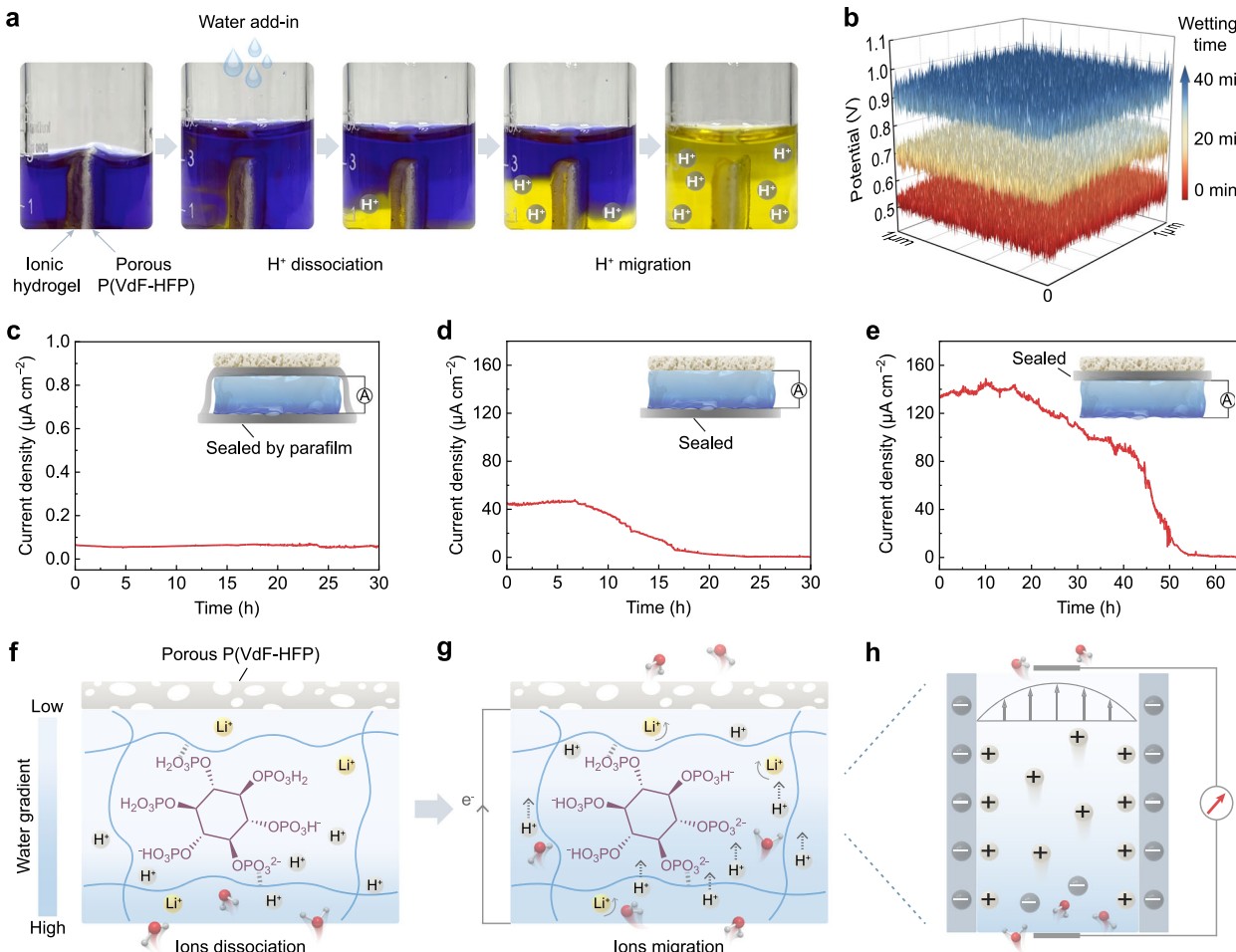

**Fig. 4 | Mechanism for moisture-electric generation of the PP/IH. a** Photos of the experiment for visualizing the dynamic change of the ion migration. The bromophenol blue/isopropanol solution (0.1% w/v) in the beaker changes color from blue to yellow corresponding to the pH changing from 4.6 to 3.0. **b** The potential variation on one side of the ionic hydrogel obtained through the KPFM test after 20 min and 40 min, while the other side was wetted. Current output of the devices with **c** both the hygroscopic and hydrophobic sides sealed, **d** only the hygroscopic side sealed, and **e** only the hydrophobic side sealed (25 °C, 50% RH). The insets show the corresponding devices with different treatments. **f**, **g**, **h** Schematic illustrating the mechanism of moisture-electric generation process in the PP/IH. **f** The incoming moisture leads to the ionization of functional groups in the hydrogel, resulting in the release of ions. **g** The water-coupled ions migrate to the hydrophobic side leaving the negatively charged hydrogel network. **h** The selective ion transportation driven by directed water flow.

Moreover, Kelvin probe force microscopy (KPFM) was employed to assess the potential variations of the ionic hydrogel during moisture sorption (Supplementary Fig. 14). The results indicate that, with ion migration caused by moisture diffusion, the non-wetted side becomes increasingly positive (Fig. 4b). In contrast, the potential on the wetted side shows almost no change (Supplementary Fig. 15).

To elucidate the effect of device structure on continuous electrical output, the following tests were conducted. Sealing both hygroscopic and hydrophobic sides with parafilm eliminates the moisture gradient and impedes directed water flow, resulting in nearly zero current output (Fig. 4c). When only the hygroscopic side is sealed, the device struggles to sorb moisture, with minimal residual water in the hydrogel migrating upward and desorbing. The difficulty in maintaining directed water flow leads to gradually diminishing current output (Fig. 4d). As shown in Fig. 4e, the device with only the hydrophobic side sealed initially exhibits an output comparable to the previous test results in Fig. 3a. However, as moisture sorption saturates, the current declines after 20 h, approaching zero at around 2 days later. Comparison of sorption mass changes between the device with the hydrophobic side sealed and untreated device also reveals that the untreated device takes longer to reach moisture saturation (Supplementary Fig. 16). The extended maintenance of moisture gradient and

directed water flow, facilitated by the vapor permeability of the porous P(VdF-HFP) and its significant hygroscopic-hydrophobic contrast with the hydrogel, enables long-term electrical output in the dynamic equilibrium of sorption and desorption.

Based on all these findings, the working mechanism of moisture-electric generation in the PP/IH can be explained as follows. Its self-sustaining electrical output relies on the continuous water flow driven by simultaneous sorption and desorption, while the enhanced power output relies on improving the moisture sorption capacity and ionic transport rate. Once exposed to air, the LiCl in the hydrogel, acting as an efficient ionic hygroscopic agent, captures moisture from the surroundings and stores liquid water within the hydrogel network (Fig. 4f). The chemical potential energy released during the transition of gaseous water molecules to the adsorbed state drives the pronounced dissociation of $H^+$ ions, leaving behind the immobilized negatively charged polymer skeletons. The strong hydration of $Li^+$ weakens the hydrogen bonds within the hydrogel[49], expanding the pathways for ion transport. Driven by ion concentration gradient, the ions undergo rapid directed migration, engendering significant charge separation (Fig. 4g). Under the dynamic equilibrium of moisture sorption and desorption, a continuous directed water flow drives dissociated ions (primarily $H^+$ ions, Supplementary Fig. 17) through the

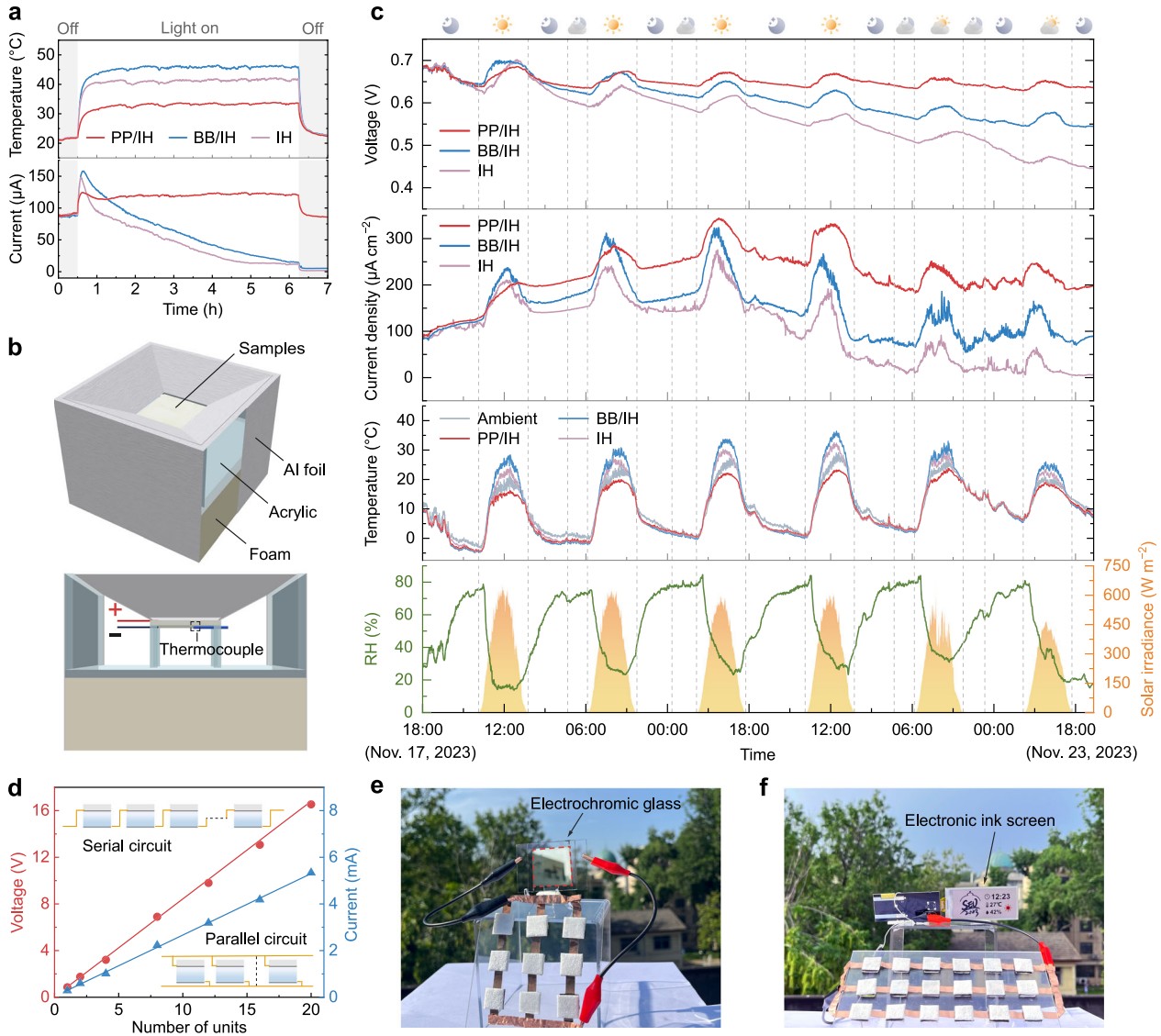

**Fig. 5 | Electrical output performance of the PP/IH in outdoor environment.**
**a** The electrical output and temperature of the PP/IH, BB/IH, and IH devices in response to on/off switching of simulated one sun (1 kW m⁻²) illumination. The test was carried out at a temperature of about 20 °C and RH of about 20%. **b** Schematic of the setup used to measure electrical performance and temperature. The concave design helps mitigate the influence of wind speed on the experimental results, while the insulating foam and aluminum foil are employed to minimize thermal conduction between the environment and the samples. **c** Outdoor electrical performance test conducted from 18:00 on Nov. 17, 2023 to 22:00 on Nov. 23, 2023, in Nanjing, China. The $V_{oc}$ and $I_{sc}$ curves of the PP/IH, BB/IH, and IH devices are presented in the first and second panels, respectively. Data on devices temperatures, ambient temperature, solar radiation intensity, and ambient relative humidity were simultaneously recorded throughout the testing period. **d** The $V_{oc}$ and $I_{sc}$ of PP/IH units (1 cm²) with different serial and parallel numbers, respectively. The insets show the serial and parallel circuits. **e** Demonstration of driving electrochromic glass (2 × 2 cm) tinted by using a 3 × 3 series-parallel connection of the PP/IH units (1 cm²) as the power source. **f** The 2.9-inch electronic ink screen directly driven by the PP/IH units (1 cm²) with a 3 × 6 series-parallel connection.

abundant negatively charged nanochannels in the hydrogel (Fig. 4h). At this point, cations are attracted, forming an electrical double layer at the solid-liquid interface. The selective ion transport allows most cations and water to pass through the pores, while anions are repelled, thereby generating electricity[8,32].

**Electrical output performance in outdoor environment**
Essentially, water sorption-desorption kinetics of the device plays a crucial role in moisture-electric generation. Various environmental factors, such as solar radiation, ambient temperature, and humidity, can affect the sorption/evaporation process, consequently influencing the electrical performance. To further explore the impact of the spectral properties of the upper layer on outdoor performance, we not only compared a single-layer ionic hydrogel device (IH) but also introduced a device featuring a blackbody-like upper surface (BB/IH),

characterized by high solar absorption and high infrared emissivity (Supplementary Fig. 18). Firstly, simulated solar illumination (1 kW m⁻²) was applied to the surfaces of PP/IH, BB/IH, and IH to investigate the effects of solar absorption (Fig. 5a). Due to the strong absorption of solar radiation (Supplementary Fig. 18) accelerating the water desorption (Supplementary Fig. 19), the current output of the BB/IH and the IH exhibits a sudden increase at the beginning but quickly followed by a gradual decrease, while the porous P(VdF-HFP) significantly prolongs the water evaporation process even under high-intensity solar radiation due to its effective solar reflection capability. Therefore, the PP/IH exhibits an increased stable current output under the influence of temperature elevation and gradually returns to its initial output level after the cessation of simulated solar illumination.

To validate the outdoor electrical performance of the PP/IH, a continuous 6-day test was conducted using the setups illustrated in

Fig. 5b and Supplementary Fig. 20. The PP/IH exhibits self-sustaining and efficient power output over an extended period, irrespective of fluctuations in environmental conditions (Fig. 5c). During the daytime, the porous P(VdF-HFP) layer reduces the evaporation temperature through radiative cooling, providing appropriate mass transfer resistance. This extends the evaporation process and thus maintains the balance between water sorption and desorption, ensuring the continuous directed water/ion flow. In contrast, although the BB/IH and the IH initially demonstrate higher output driven by absorbed solar radiation, their decay becomes more pronounced as the water content and evaporation rate decrease. This is further confirmed by simulation results of the evaporative mass changes for different devices (Supplementary Fig. 21). The PP/IH maintains a relatively stable evaporative mass flux within 10 hours, while the evaporative rates of the BB/IH and the IH drop below that of the PP/IH after about 4 h, approaching zero after 8 h ($30\,°C$ and 40% RH with solar radiation of $800\,W\,m^{-2}$). The trends in evaporation flux changes for different devices show consistency with the current variations in Fig. 5a. Additionally, the porous P(VdF-HFP) emits thermal radiation to the space, effectively offsetting the exothermic heat from moisture sorption and enhancing local relative humidity during the night, facilitating water sorption and the restoration of the water gradient (Supplementary Fig. 22). The upper layer of BB/IH achieves a comparable temperature drop to the porous P(VdF-HFP) film through nighttime radiative cooling (Supplementary Fig. 23). Consequently, the electrical outputs of BB/IH and PP/IH exhibit similar trends at night. However, excessive daytime evaporation makes it challenging to recover the water gradient in BB/IH to its initial level, leading to a gradual decay in output over multiple diurnal cycles. Due to the slower nighttime regeneration of the IH, its performance degradation is more pronounced. We also investigated the moisture sorption/desorption behaviors of the PP/IH under different RH (Supplementary Fig. 24), along with the sustaining output performance in high-temperature weather with intense solar radiation (Supplementary Fig. 25). The results demonstrate that the PP/IH exhibits excellent all-weather adaptability. Furthermore, due to the combined cooling effects of evaporation and radiation, the PP/IH achieves an average sub-ambient cooling of $-6\,°C$ between 10:00 and 15:00, indicating its potential application as a cooling device.

By integrating multiple PP/IH device units, the electrical output can be easily expanded to cater to diverse application scenarios. Twenty individual units connected in series generated a linearly increasing voltage output exceeding 16 V (Fig. 5d). The current output generated by the parallel connection of 20 units is $-5.3\,mA$, showcasing the impressive scalability of the PP/IH. Arranging the PP/IH units in series and parallel combinations enables their utilization as a DC power source for electrical devices. For instance, a $3 \times 3$ series-parallel connection of the PP/IH units is sufficient to drive electrochromic glass silver electrodeposition[50], demonstrating its potential for smart window application and building energy saving (Fig. 5e). A $6 \times 3$ series-parallel integrated device is even capable of powering a 2.9-inch electronic ink screen display (Fig. 5f, Supplementary Movie 1).

## Discussion

In summary, we developed a bilayer polymer designed for efficient and stable electricity generation from the hydrological cycle through material regulation and thermal exchange with the natural environment. A PP/IH unit continuously produces a high $V_{oc}$ of $-0.88\,V$ and $I_{sc}$ of $-306\,\mu A$, with a maximum power density of $51\,\mu W\,cm^{-2}$ at $25\,°C$ and 70% RH. During a continuous 6-day outdoor test, the PP/IH exhibits exceptional environmental adaptability and sustaining electrical output. The radiative cooling effect of the top layer prevents excessive daytime evaporation of the hydrogel while facilitating nighttime moisture sorption, thereby maintaining a continuous water/ion flow. The incorporation of LiCl enhances moisture sorption and inhibits hydrogen bonds formation within the hydrogel, resulting in efficient

ion dissociation and rapid ion transport in the PP/IH. Integration of multiple PP/IH units can provide power sources for potential applications in different scenarios such as building energy conservation and self-powered wearable devices. This work provides insights for improving moisture-electric generation performance in hot and dry regions where conventional generators are often unsuitable and advancing the development of eco-friendly energy systems.

## Methods

### Materials

Poly(vinyl alcohol) (PVA, degree of hydrolysis: 98-99%, degree of polymerization: 2400–2500), phytic acid (PA, 50 wt% in $H_2O$), lithium chloride (99.9%) and bromophenol blue indicator (0.1% w/v in isopropyl alcohol) were purchased from Aladdin Co., Ltd. Poly(vinylidene fluoride-co-hexafluoropropene) [P(VdF-HFP)] and acetone (99%) were purchased from Sigma-Aldrich Co., Ltd. Carbon cloth (hydrophilic and hydrophobic) and iron foam were purchased from Guangdong Canrd New Energy Technology Co., Ltd. All reagents were used without further purification.

### Preparation of the porous P(VdF-HFP) layer

The porous P(VdF-HFP) layer was prepared by phase inversion-based method[24,34]. First, P(VdF-HFP) powder was dissolved in acetone, followed by the addition of DI water, to prepare a precursor solution of P(VdF-HFP), acetone, and water with a mass ratio of 1:8:1. Next, iron foam was thoroughly immersed in the precursor solution and left to dry completely. Subsequently, it was soaked in a $1\,mol\,L^{-1}$ hydrochloric acid solution to dissolve the iron foam. Finally, the obtained porous P(VdF-HFP) film was subjected to ultrasonic cleaning in DI water and then dried. Besides, to modify the spectral properties of the upper layer to approximate those of a blackbody, acrylic lacquer black paint was sprayed onto the existing P(VdF-HFP) film and dried.

### Preparation of the ionic hydrogel layer and integrated devices

The ionic hydrogel layer was synthesized by a simple one-pot method. Firstly, 4.5 g LiCl was added into 12.5 ml DI water and stirred until completely dissolved. Then 3 g PVA and 7.5 ml PA were added into above solution and dissolved at $95\,°C$ for 3 h to obtain a homogeneous solution. The mixed solution was subjected to ultrasonic deaeration, followed by pouring into molds of different thicknesses at $-15\,°C$ overnight to obtain PVA-PA-LiCl hydrogel. Hydrophilic and hydrophobic conductive carbon cloth were connected to the respective sides of the hydrogel to serve as electrodes. Afterwards, the porous P(VdF-HFP) layer and the ionic hydrogel layer were assembled under pressure, with the side covered with hydrophilic carbon cloth facing outward. As control groups in the experiment, devices based on PVA, PVA-LiCl, and PVA-PA hydrogels were prepared using a similar procedure.

### Characterization

The morphology of dried ionic hydrogel and porous P(VdF-HFP) film were characterized by scanning electron microscope (SEM, Zeiss Sigma 300, Germany). Energy-dispersive X-ray spectroscopy (EDS) in SEM was employed to study the elemental distribution of the hydrogel. The solar reflectance and absorptance were measured using a UV−vis−NIR spectrophotometer (Shimadzu UV-3600i Plus, Japan) equipped with an integrating sphere. The infrared emissivity was collected by a FTIR spectrometer (Thermo Fisher Nicolet iS50, USA) attached to a gold integrating sphere. The pore sizes distribution of porous P(VdF-HFP) layer was analyzed by a mercury intrusion porosimetry (Micromeritics Autopore V9620, USA). The chemical structures of hydrogels were characterized by FTIR spectroscopy (Thermo Fisher Nicolet iS20, USA) in attenuated total-reflection (ATR) model. The XRD patterns were measured using an X-ray diffractometer (Rigaku SmartLab SE, Japan). The ionic conductivity was determined through

electrochemical impindence spectroscopy (EIS) measurement using an electrochemical workstation (CH Instruments CHI660D, USA). The potential variation of the ionic hydrogel was obtained from KPFM (Bruker Dimension Icon, Germany).

## Electrical performance measurements

Temperatures monitored by T-type thermocouples and electric signals were recorded by a data acquisition system (Keysight DAQ970A, USA). The relative humidity was monitored using a hydrometer (Shandong Renke COS-03-2WD, China). The indoor electrical output testing was conducted in a constant temperature and humidity chamber (Shanghai Yiheng LHS-250HC, China). All other indoor electrical output performance tests were conducted at 25 °C and 75% RH unless otherwise specified. Mass variations of ionic hydrogels were measured by an electronic balance (Ohaus PR224ZH, USA). The weather parameters during the outdoor experiments were recorded using a digital high-precision weather station.

## Data availability

The data supporting the findings of the study are included in the main text and supplementary information files. Raw data can be obtained from the corresponding author upon request. Source data are provided with this paper.

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

## Acknowledgements

This work was supported by the National Natural Science Foundation of China (No. 52276178, D.Z.), Jiangsu Province Carbon Peak and Carbon Neutrality Science and Technology Innovation Special Fund (No. BE2023854), and Southeast University Interdisciplinary Research Program for Young Scholars.

## Author contributions

D.Z., T.L., and C.G. proposed the concept and designed the experiments. C.G. and H.T. conducted the experiments. T.L. and P.W. gave advice on experiments. C.G., Q.X., H.P., X.Z., and F.F. performed the characterizations. C.G. and H.T. performed the simulations and theoretical calculations. C.G. and D.Z. prepared the manuscript and contributed to the interpretation of the results.

## Competing interests

The authors declare no competing interests.
