## [Peer Review File · Nature Communications]

Radiative cooling assisted self-sustaining and highly efficient moisture energy harvestingEditorial Note: This manuscript has been previously reviewed at another journal that is not operating a transparent peer review scheme. This document only contains reviewer comments and rebuttal letters for versions considered at *Nature Communications*. Mentions of the other journal have been redacted.

REVIEWER COMMENTS

Reviewer #1 (Remarks to the Author):

According to the detailed response by the authors, the concerns on the novelty of this work has been addressed to some extent. It is suggested to supplement some related references combining radiative cooling with sustainable energy harvesting technology based on moisture-electric generator, triboelectric generator or thermoelectric generator (e.g., Adv. Sci. 2023, 10, 2206925. DOI:10.1002/advs.202206925; Wang, Si, et al. A Versatile Strategy for Concurrent Passive Daytime Radiative Cooling and Sustainable Energy Harvesting. Small 20.6 (2024): 2305706; Zhang, JH, et al. Versatile self-assembled electrospun micropylam arrays for high-performance on-skin devices with minimal sensory interference. Nat Commun 13, 5839 (2022); Ishii, Satoshi, et al. Transparent thermoelectric device for simultaneously harvesting radiative cooling and solar heating. Materials Today (2024)), which is significant to summarize the recent research progresses on the high efficiency energy harvesting enabled by radiative cooling and different energy harvesting methods. Based on the reported progresses, point out the importance of the proposition of new mechanism for this field (e.g., the influence of radiative cooling on the kinetics of moisture sorption/desorption proposed in this work).

Reviewer #2 (Remarks to the Author):

The authors have addressed all my concerns well. The paper can be accepted for publication.

Reviewer #3 (Remarks to the Author):

I reviewed this work before, which was submitted to [REDACTED]. In that time, I gave positive comments. However, this work was rejected unfortunately. After carefully checking the response letter, I think the authors have addressed my concerns. Overall, this work is suitable for publishing in Nature Communications.

Point-by-point response to the reviewers' comments

Dear Reviewers:

We are deeply grateful for your comprehensive and insightful feedback on our manuscript titled “**Radiative Cooling Assisted Self-Sustaining and Highly Efficient Moisture Energy Harvesting**”. Your detailed comments and constructive suggestions have been invaluable in guiding our revisions and enhancing the overall quality of our work. Thank you for your dedication and expertise, which have significantly contributed to the improvement of our manuscript. We have made the necessary revisions to address your concerns. The revised contents were highlighted **in red color** in the manuscript. Our detailed responses are as follows:

Reviewer #1:

According to the detailed response by the authors, the concerns on the novelty of this work has been addressed to some extent. It is suggested to supplement some related references combining radiative cooling with sustainable energy harvesting technology based on moisture-electric generator, triboelectric generator or thermoelectric generator (e.g., Adv. Sci. 2023, 10, 2206925. DOI:10.1002/advs.202206925; Wang, Si, et al. A Versatile Strategy for Concurrent Passive Daytime Radiative Cooling and Sustainable Energy Harvesting. Small 20.6 (2024): 2305706; Zhang, JH, et al. Versatile self-assembled electrospun micropylamid arrays for high-performance on-skin devices with minimal sensory interference. Nat Commun 13, 5839 (2022); Ishii, Satoshi, et al. Transparent thermoelectric device for simultaneously harvesting radiative cooling and solar heating. Materials Today (2024)), which is significant to summarize the recent research progresses on the high efficiency energy harvesting enabled by radiative cooling and different energy harvesting methods. Based on the reported progresses, point out the importance of the proposition of new mechanism for this field (e.g., the influence of radiative cooling on the kinetics of moisture sorption/desorption proposed in this work).

Response: Thank you for your meticulous review of our manuscript. Your valuable insights are greatly appreciated. We have carefully considered all your comments and made the

necessary revisions. We have revised the Introduction part to briefly summarize recent research integrating radiative cooling with sustainable energy harvesting technologies and added relevant references. Additionally, we further clarified the novelty of this work in terms of its concept and working mechanism. We hope these revisions address your concerns and improve the overall quality of this work.

For your convenience, the revised content with highlight on **page 4, lines 55–70 in the main text** is attached as below:

“Daytime radiative cooling achieves passive cooling without energy consumption during the whole diurnal cycle by reflecting sunlight and emitting mid-infrared thermal radiation to the space²²⁻²⁶. Recent studies have integrated radiative cooling with various sustainable electricity harvesting technologies. For example, radiative cooling films-based triboelectric nanogenerators have been used for personal thermal management and biomechanical energy harvesting^{27,28}. Radiative cooling has also been employed to increase the temperature difference between the hot and cold sides of thermoelectric generators, thereby improving power output performance under solar irradiation^{29,30}. Additionally, an asymmetric bilayer cellulose-based fabric that is capable of independent radiative cooling and transpiration-driven electricity generation has been developed³¹. In this work, we aim to leverage the influence of radiative cooling process on moisture sorption-desorption kinetics and propose an efficient and sustainable moisture-electric generation solution from a thermodynamic perspective by leveraging the external environment. By strategically introducing radiative cooling to improve thermal management between materials and the environment, it holds promise for constructing a long-term hydrological cycle to maintain directed water/ion flow within the material under real-world environment fluctuations.”

Added references:

- 27 Zhang, J. H. *et al.* Versatile self-assembled electrospun micropylamids for high-performance on-skin devices with minimal sensory interference. *Nat. Commun.* **13**, 5839 (2022).

- 28 Wang, S. *et al.* A versatile strategy for concurrent passive daytime radiative cooling and sustainable energy harvesting. *Small* **20**, e2305706 (2024).
- 29 Ren, W. *et al.* High-performance wearable thermoelectric generator with self-healing, recycling, and lego-like reconfiguring capabilities. *Sci. Adv.* **7**, eabe0586 (2021).
- 30 Ishii, S., Bourgès, C., Tanjaya, N. K. & Mori, T. Transparent thermoelectric device for simultaneously harvesting radiative cooling and solar heating. *Mater. Today* (2024). <https://doi.org/10.1016/j.mattod.2024.03.012>
- 31 Zhang, K. *et al.* A zero-energy, zero-emission air conditioning fabric. *Adv. Sci.* **10**, e2206925 (2023).

Reviewer #2:

The authors have addressed all my concerns well. The paper can be accepted for publication.

Response: We are immensely grateful for your thorough review and the valuable feedback. Your insightful comments and suggestions significantly enhanced the quality of our work. We deeply appreciate the time and effort you invested in evaluating our submission.

Reviewer #3:

I reviewed this work before, which was submitted to [REDACTED]. In that time, I gave positive comments. However, this work was rejected unfortunately. After carefully checking the response letter, I think the authors have addressed my concerns. Overall, this work is suitable for publishing in Nature Communications.

Response: Thank you once again for your support and encouragement throughout the review process. Your expertise and constructive critique have been instrumental in refining our research. We are sincerely grateful for your recommendation of acceptance.

REVIEWERS' COMMENTS

Reviewer #1 (Remarks to the Author):

The revised version has solved the reviewers' concerns and the manuscript quality has been greatly improved. I suggest it can be accepted now.

Point-by-point response to the reviewers' comments

Reviewer #1:

The revised version has solved the reviewers' concerns and the manuscript quality has been greatly improved. I suggest it can be accepted now.

Response: We are sincerely grateful for your recommendation of acceptance. Your insightful comments and suggestions significantly enhanced the quality of our work.